# A Qualitative Analysis of Rural Community Vaccination Barriers During the COVID-19 Pandemic

**DOI:** 10.3390/vaccines12121442

**Published:** 2024-12-21

**Authors:** Kimberly C. McKeirnan, Megan R. Undeberg, Skylar Zelenko, Ghazal Meratnia

**Affiliations:** College of Pharmacy and Pharmaceutical Sciences, Washington State University, Spokane, WA 99202, USA; meganru@wsu.edu (M.R.U.); skylar.zelenko@wsu.edu (S.Z.); ghazal.meratnia@wsu.edu (G.M.)

**Keywords:** rural health, university extension, COVID-19, vaccine uptake, qualitative research

## Abstract

Background/Objectives: Rural communities in the United States experience increased disparity of care for both general healthcare services and access to routine vaccines. Previous research has indicated a 40% lower vaccination rate in rural communities, as compared to urban counterparts. Having a better understanding regarding factors influencing lower vaccination rates in rural areas could help public health officials prepare for future vaccination efforts. This research sought to gather and evaluate the opinions of people who live and work in rural areas regarding barriers to COVID-19 vaccine uptake. Methods: A semi-structured qualitative key informant interview design was utilized by researchers to gather opinions from university Extension staff in Washington State. Interview transcripts were analyzed using the Theory of Planned Behavior (ToPB) framework to evaluate COVID-19 vaccination-related intentions and motivational factors that the Extension staff observed among rural populations in their communities. Results: Twenty-one participants representing 34 out of the 40 Extension offices responded and were interviewed during fall 2023. Using the ToPB constructs, nine barriers were identified. Attitude-related barriers included the following: inherent social distancing in rural location negating vaccine necessity; lack of early vaccine availability in rural locales; concerns regarding ineffectiveness of the vaccine; and inadequate dissemination of vaccine information to non-English language speakers and those with limited access to technology. Subjective norm barriers included the following: perception of exclusion of rural populations’ unique needs during design and implementation of vaccine mandates; exertion of social pressures on rural individuals’ vaccine uptake decision; and highly visible breakdown in standard trust in core community institutions and leadership. Barriers related to loss of perceived behavioral control included vaccine mandates impacting self-perceived loss of autonomy and limitations in vaccine technology information impacting perception of vaccine safety. Conclusions: By identifying barriers to vaccination in rural communities during the COVID-19 pandemic, future outreach efforts can be designed to improve intention and lead to stronger vaccination uptake.

## 1. Introduction

Despite wide vaccine availability, outbreaks of vaccine-preventable diseases continue to occur. Rural areas in the United States experience disparities in access to routine vaccines related to a general lack of healthcare services [1,2]. Previous rural research has indicated a 40% lower mean vaccination rate in rural communities, as compared to their urban counterparts [2]. The Coronavirus Disease 2019 (COVID-19) pandemic further exposed this disparity. In the first year that COVID-19 vaccines were available, only 59% of rural county residents in the United States had received their first dose as compared to 74% of patients from urban counties [3].

The issue of lower vaccination rates in rural areas compared to urban areas is complex and multifaceted. There are several proposed theories attempting to explain this phenomenon; one of the predominant theories is limited access to healthcare services. According to the Centers for Disease Control and Prevention (CDC), an analysis using a six-level population density classification found that a larger proportion of people living in rural and suburban areas had to travel to nonadjacent counties that were farther from their county of residence for vaccination (13.9% and 14.6%, respectively) compared with those in the most urban counties (10.3%) [3]. The CDC also created the COVID-19 Vaccination Field Guide to address low vaccination rates in rural areas [4]. This document encourages utilization of mobile health units and home visits to address limited care access [4].

Another hypothesis is that social vulnerability may be the culprit for low rural patient vaccination rates. A CDC report using a social vulnerability index identified 15 factors such as poverty and lack of transportation that decreased community capacity to respond in an emergency [5]. Disparities in county-level vaccination coverage by social vulnerability have increased as vaccine eligibility has expanded, especially in areas outside of large cities [5]. In May 2021, only 40.6% of people in the most vulnerable counties were vaccinated against COVID-19, compared with 52.9% among the least vulnerable [5]. Vaccination coverage among adults is also reduced among people in counties with lower socioeconomic status (44.3% vs. 62%) and with higher percentages of households with children, single parents, and persons with disabilities (42% vs. 60.1%) [6].

In addition to socioeconomic status, racial and ethnic disparities also appear to be a factor. Concerns over historical medical mistreatment, unethical medical research, and ongoing healthcare inequities, have evolved into mistrust and skepticism regarding even ethnically and racially targeted vaccination campaigns [7]. A 2023 systematic review found that concerns about vaccine safety, vaccine effectiveness, and side effects contributed to COVID-19 vaccine hesitancy among minority groups [8]. Additionally, underrepresentation of minority groups in the COVID-19 vaccine trials was found to contribute to vaccine hesitancy and reduced uptake among minority populations [8]. Even before the pandemic, routine vaccination rates varied depending on race, ethnicity, household income, education level, and insurance status [9]. Notably, seasonal influenza vaccine rates have also varied between White, Black, Hispanic, and Asian populations (72%, 61%, 64%, and 71%, respectively) in older adults [9]. In a comparison of younger adults between the ages of 18 and 64, seasonal influenza vaccination rates also varied between Whites (42%), Blacks (35%), and Hispanics (35%) [9].

Higher rates of vaccine hesitancy among rural populations likely also contributes to lower immunization rates. The Kaiser Family Foundation reports that rural residents (35%) are among the most vaccine-hesitant groups, along with Republicans (42%), individuals 30–49 years old (36%), and Black adults (35%), compared with the reported vaccine hesitancy rate of 27% among the general public [10]. Individuals living in rural areas in the U.S. were significantly less likely to be vaccinated against COVID-19 than individuals living in suburban and urban north America. This phenomenon was predicted even before COVID-19 vaccines were available. In a 2020 survey, 31% of people in rural areas said they would “definitely get” the vaccine, compared to 42% of people in urban areas and 43% of people living in suburban areas [10].

Having a better understanding regarding lower vaccination rates in rural areas could help government and public health officials address vaccine hesitancy and be more prepared to address future pandemics. However, it can be difficult for people outside of rural communities to fully understand the challenges faced by rural citizens, particularly since these issues seem to have many variables. University Extension staff have in-depth experience offering educational and outreach services in their communities. They work, and often live, in the rural community they serve and have longstanding relationships with community organizations and individuals. This research sought to gather and evaluate the opinions of people who live and work in rural areas regarding barriers to COVID-19 vaccine uptake.

## 2. Materials and Methods

### 2.1. Study Design

The Consolidated Criteria for Reporting Qualitative Studies (COREQ) guidelines were utilized during project design [11]. Methods used in this study were found to satisfy the criteria for Exempt Research by the Washington State University (WSU) Human Research Protection Program (Institutional Review Board 20129-001).

A semi-structured qualitative key informant interview design [12,13] was utilized by researchers to gather opinions from university Extension staff in Washington State to evaluate barriers to COVID-19 vaccine uptake in rural areas. Key informant interviews are in-depth conversations where experts provide detailed insight into a problem or topic [14]. In this case, Extension staff are ideal key informants because they possess comprehensive knowledge of the challenges and opportunities in the communities they serve. Each Extension office serves a rural county population with unique culture, demographics, opportunities, and limitations. Since WSU has at least one staff member in each county in Washington State and an additional site serving the Colville Nation tribal community, key informant interview methods were selected to gather insight into COVID-19 vaccination barriers representing rural inhabitants of the entire state of Washington.

### 2.2. Study Participants

University Extension staff have in-depth experience offering educational and outreach services in their communities. Extension staff, as community experts, can serve as a collective set of eyes and ears in providing insight to the challenges that residents in Washington State’s rural and underserved areas are facing in accessing vaccines and vaccine services. WSU Extension offices are located in all 39 counties in Washington State, with one additional office on the land of the Colville Reservation [15]. Washington State, located in the Pacific Northwest of the United States, is the 20th largest U.S. state by area and the 13th by population [16].

In 1862, the Morrill Land Grant College Act was signed into legislation by President Abraham Lincoln, setting aside federal land to create at least one public educational institution, known as a land grant university, in each U.S. state to provide fair education for all [17,18]. Cooperative Extension Services, hereafter referred to as Extension, were established in 1914 when the Smith–Lever Act enabled these universities to share crop and animal production information with the public [18]. Over time, as the demographics of American citizens have shifted from rural industries into urban and suburban populations, Extension services have shifted their focus as well to reflect differing needs. While today’s Extension may continue to support rural populations’ crop innovations, animal husbandry, and other rural industries, they may also facilitate leadership training, nutrition skills, and youth mentorship throughout the United States. Extension personnel and services are located in or nearby each of the nearly 3000 counties across the United States [19].

Washington’s land grant institution, Washington State University, provides Extension services and affiliated personnel “to provide solutions to local problems and stimulate local economies” [15]. Extension staff, located in each county, have a unique perspective of those communities they interface with: community-based liaisons, specific demographics, inherent needs, and engagement with the spectrum of ethnicities, races, and age groups from birth to death. Additionally, WSU Extension staff are, by nature, integrated into the counties they serve with the goal of “creating and delivering targeted research-based knowledge and education” [15].

Each Extension office was contacted individually via email using the contact email address listed on the Extension office website. A recruitment email was sent by the primary investigator (KM) and contained IRB-approved language, including consent and confidentiality information, and offering the participant a $10 gift card as an incentive and in appreciation for their time participating in the interview. Potential participants were requested to reply to the email if they were willing to participate in the interview and to suggest another member of their Extension office staff who may be willing to participate if they were not interested. Once a response was received by the PI, a Zoom (Zoom Communications, Qumu Corporation, San Jose, CA, USA) interview was scheduled at a time that was convenient for the participant. If no response was received, a reminder email was sent one week later. No further correspondence was sent if a reply was not received to the two emails. Due to the diverse nature of the inhabitants of the state of Washington, recruitment emails were sent to representatives of all 40 WSU Extension offices.

### 2.3. Interview Script Development

A semi-structured key informant interview was developed, including a combination of open-ended and closed-ended questions. Closed-ended questions were used to gather demographic data. The open-ended questions were utilized to elicit more detailed and thoughtful responses regarding COVID-19 vaccination barriers. A semi-structured format was chosen so that the interviewer would use the script as a guide while having the freedom of including probing questions when needed to dive into topics deeper. Key informant interview methodology allows for variation from the script to add probing questions as needed to gain additional insight and clarification on an individual basis. After development, the interview script was piloted by two of the Doctor of Pharmacy students and adjustments were made as needed to refine the script language.

### 2.4. Conducting Interviews

The interviews were conducted via Zoom by two members of the research team. The primary interviewer for each interview was a student pharmacist. Students were chosen to conduct the interviews because they were not employed by the university and the research team believed having student interviewers would make the participants more comfortable and willing to share sensitive information. One student (GM) was in her final year of PharmD training and had previous qualitative research experience during training for a Master’s degree in public health. The other student (SZ) was in her third professional year of the PharmD program. Student researchers were trained by faculty researchers (KM and MU) prior to conducting the interviews. During each interview, a faculty researcher was present in the Zoom meeting to answer study-related questions and ensure technology was working properly but had minimal participation. The interviewers had not met the interview participant previously and did not have any prior relationship with any of the participants. In two instances, one faculty researcher (KM) had previously met an Extension staff member who was to be interviewed, so instead the other faculty researcher (MU) attended the interviews.

At the beginning of the interview, participants were informed that the interview would take 20–30 min, participation was voluntary, and they could end the interview at any time or decline to answer any of the individual questions. The participants were also informed that all identifying information would be redacted from the transcripts before initiating the data analysis. Anonymizing participant information and ensuring there were no prior connections between the participants and interviewers was done by design to encourage Extension staff to express personal opinions and describe experiences without concern of offense or repercussions from negative comments. The Zoom transcription feature was used for producing a written documentation of the interviews. Transcriptions and corresponding audio files were reviewed by KM to check for transcription quality and correctness. Errors identified in transcriptions were corrected by listening to and manually transcribing the audio file where needed.

### 2.5. Data Analysis

After transcriptions were de-identified and errors removed, the transcriptions were sent to the remaining researchers for thematic coding. Deductive qualitative coding methods were used [20]. The research team met to train the student researchers on the coding process and to practice coding one transcript together. The researchers performed first-level coding on the remaining transcripts independently to reduce bias. First-level coding is a systematic method used to identify relevant and repeated concepts to form codes, in this case identifying parallel responses from participants regarding similarities and differences of the experiences with COVID-19 vaccinations in their community. After first-level coding was completed, the researchers met to perform second-level coding, which involved combining codes using the constructs of the theoretical framework to identify specific barriers. Pertinent quotes that align with the identified barriers were also identified. Results were then reviewed and discussed by the researchers to confirm proper categorization. Disagreement among the researchers about categorization was minimal and was resolved by further group discussions.

### 2.6. Theoretical Framework

The Theory of Planned Behavior (ToPB) is guided by the principle that human behavior is comprised of three types of considerations: behavioral beliefs, which produce attitude toward the behavior; normative beliefs, or beliefs about the expectations of others, which create subjective norms; and control beliefs, which include perceptions about factors that lead to or prohibit behavior, also known as perceived behavioral control [21,22]. This widely known framework has been applied to more than 4200 published manuscripts in a variety of social and behavioral science fields [22]. In this case, researchers attempted to utilize the ToPB framework to organize and evaluate opinions gathered from university Extension staff in Washington State regarding barriers of COVID-19 vaccination uptake during the pandemic. This framework was chosen because it incorporates both social influences and personal factors as behavioral predictors.

## 3. Results

Twenty-one participants representing 34 out of the 40 Extension offices responded to the request and were interviewed during fall 2023. The interviews were continued with the purpose of gathering information until saturation was met, meaning subsequent interviews did not add any novel information [23]. Meeting saturation is utilized in qualitative research rather than power analyses. After reviewing the 21 transcripts, the researchers agreed that saturation had been met and interviewing additional Extension staff would be unlikely to identify new information [23].

Using the ToPB constructs, nine barriers were identified as impacting rural populations’ decision-making process of ultimately receiving or not receiving COVID-19 vaccination. These included four barriers related to the construct of Attitudes, three related to Subjective Norms, and two impacting Perceived Behavioral Control [21]. Barriers organized using the constructs of the ToPB are shown in Figure 1.

### 3.1. Demographics

Demographics of participants are included in Table 1.

### 3.2. Attitudes

Attitudes are an individual’s personal beliefs about the outcome of the behavior. As an individual considers whether to perform a behavior, they are reflecting on the positive or negative outcomes of performing that behavior and whether those outcomes are worthwhile given the effort and resources involved in performing it. In this case, information was gathered about factors that indicate whether getting vaccinated was viewed as a worthwhile outcome in communities observed by study participants. The following barriers to COVID-19 vaccination uptake were identified as attitudes held by people living in rural communities who were hesitant or chose not to be vaccinated.

#### 3.2.1. Barrier 1: Point of View That Rural Residency Provides Inherent Social Distancing and Protection from the Spread of COVID-19, So Vaccination Is Not Necessary

There was lower vaccination uptake in rural areas early in the pandemic because of the belief among rural populations that COVID-19 was less likely to spread and infect rural populations. Several factors may have contributed to this, including less frequent contact with others due to the nature of rural work. Participants reported observing in their communities the widespread belief that being in a rural area provided relief from the spread of COVID, as described in the following illustrative quotes:•“I’m out in middle nowhere. COVID is not going to reach across 10 miles and infect me” (Participant 20)•“I heard a lot of conversation around not needing to be vaccinated because they are farming outdoors and didn’t have to be in close contact with people” (Participant 9)•“This is more of a working-class area and health care is more [treatment-based] care instead of preventative medicine care. We’re in logging and fishing communities that get medical care when injured. [Local people reported] ‘I’m not sick so I don’t understand what’s going on here’” (Participant 18)

#### 3.2.2. Barrier 2: Perception That if COVID-19 Vaccination Was Valued and Essential to Rural Public Health, More Vaccines Options Would Have Been Available and Sooner

Perceptions in rural communities that getting vaccinated against COVID-19 was not important were amplified when vaccines were made available in urban portions of counties before the rural areas:•“Particularly it seemed like [name of large city] had first access to the vaccines and [name of large company] had a lot of the vaccine” (Participant 9)•“One thing that was somewhat of a barrier the different kinds of vaccines. There were Johnson & Johnson vaccines for a while, and then Moderna and Pfizer. At first folks just had access to Johnson & Johnson out here in our rural area, and it was frustrating for some people that we didn’t have the same variety available at the beginning that other places had, the different vaccinations that they had in more urban areas. That was tough in some of our rural areas” (Participant 11)

#### 3.2.3. Barrier 3: Belief That the COVID-19 Vaccine Is Ineffective

Interview participants observed feelings of hesitancy among people living in rural communities about the COVID-19 vaccination stemming from the belief that the vaccine was not effective at preventing the spread of the disease. Participants also reported that people living in rural communities believed that if they did become ill with COVID-19, the illness would not lead to serious medical problems, as expressed in the following quotes:•“Some people said, well I never got vaccinated and I never got COVID, so what was the big deal? Other people went through the whole vaccine regime and still got it, sometimes more than once” (Participant 2)•“I think there’s two camps: the camp who have gotten vaccine because they believe it’s the right thing to do and they want to keep family safe and friends safe and all that type of stuff. And then you have the other camp who believe ‘I didn’t get vaccinated and I’m doing just fine’ too. You can’t say they’re wrong, they just skirted [COVID-19] or they got sick but didn’t get that sick. But then there are people that we all know that did pass away from [COVID-19] in our communities. So, that component is hard to justify as well” (Participant 18)•“It was challenging for people to have faith that what they were hearing was accurate. There was uneasiness about it” (Participant 11)

#### 3.2.4. Barrier 4: Insight That COVID-19 Vaccine Information Was Inadequately Disseminated to Non-English Speaking Populations and Those with Limited Access to Technology

Another factor reported by participants that contributed to vaccine hesitancy was limited information. Participants reported observing that, at times, information was not communicated in a way that could be understood by people who did not speak English or did not have ready access to a computer.
•“There was very limited information for Spanish-speaking people. It was terrible. They were frightened. They didn’t know what they were getting into, what was happening” (Participant 2)•“Among some of the poorer economic demographics in our community, I saw that they were not getting enough information to make informed decisions so that they just weren’t making decisions. A lot of folks said, ‘I don’t know what to do, so I’m going to do nothing’” (Participant 9)•“The cultures that are not the dominant cultures anywhere have a distrust or hesitancy to trust the mainstream culture, government, medicine, everything that they see. So I would guess that the minority populations in any county were less likely to be vaccinated than the mainstream culture” (Participant 20)

### 3.3. Subjective Norms

Subjective norms are individual perceptions about social pressures to perform the behavior, in this case pressure for and against getting vaccinated. Participants identified mistrust of government, pressure from political parties, content shared on social media, and social pressure from friends and family as subjective norms that influenced their decisions about getting vaccinated against COVID-19.

#### 3.3.1. Barrier 5: Perception That COVID-19 Vaccine Rules and Regulations Were Made to Address the Needs of Urban Populations Without Input and Consideration of Rural Community Attributes

Participants reported that people living in rural communities felt that state rules about shutdowns and vaccine mandates only considered those living in crowded urban areas and were not necessary in rural areas. In Washington State on 9 August 2021, Governor Jay Inslee issued Proclamation 21–14, thereby requiring “all healthcare providers (this includes all employees, contractors, volunteers, and providers of goods and services who work in a health care setting) to be fully vaccinated against COVID-19 by 18 October 2021” [24]. Many local organizations and businesses also adopted policies requiring people to be vaccinated to utilize local non-health-related services. One common response from key informants was the belief among community members that rural areas where people live far apart and often interact outdoors should not be subject to the same rules that were applied to Washingtonians living in dense urban areas, as shown in the following quotes:•“There were a lot of I think feelings of unfairness that the rules were made for people in massive urban areas where there was so much risk of exposure and yet they were applied to people in smaller communities where the risk of exposure was vastly lower” (Participant 3)•“[I observed local people] blaming our misfortunes on decisions that are made by folks that don’t represent us. It doesn’t help that the Governor is a [urban area] resident who was making decisions from the top down. Everybody in the state was having to follow his mandates, and for those of us living in [rural area] it made it even worse that he didn’t seem to understand us” (Participant 20)•“I think that public officials can’t assume people will trust them or listen to them. I think they should actually work as liaisons with trusted community leaders, like nonprofits, public service spaces, church leaders” (Participant 19)

Extension staff expressed that people living in rural communities were skeptical that the government understood their concerns or had their best interests in mind. Aside from state government COVID-19 regulations, key informants also expressed the observation that there was a level of mistrust of the government in general among local community members. This observation is shared in the following illustrative quotes:•“I think mistrust of government. Certain individuals mistrust anything the government does” (Participant 8)•“If you look at the people that didn’t get vaccinated, it’s very much a ‘don’t tread on me,’ ‘the government can’t advise me,’ and ‘I don’t trust government,’ mentality” (Participant 21)•“I think there’s a healthy skepticism of government in my area and the state government specifically” (Participant 18)

#### 3.3.2. Barrier 6: Social Pressures Exerted Substantial Influence on COVID-19 Vaccination Decisions

External social pressures, such as those from political parties and social media, exerted substantial influence on those who chose not to be vaccinated against COVID-19 in rural areas. Interview participants reported that COVID-19 vaccination became a polarizing issue, as described in the following quotes:•“I would say politics was the number one reason why people weren’t [getting vaccinated]” (Participant 16)•“Just political polarization. Pretty intense. Rural versus urban. Conservative versus liberal” (Participant 10)•“The national news turned [getting vaccinated] into a political football and we didn’t get much out of that” (Participant 6)

The influence of social pressure from the community and information on social media also acted as a perceived behavioral norm for people in rural areas. Extension staff reported observing the influence of content provided through social media as being very influential.
•“We had widespread rumors going around communities because people were just talking but didn’t think critically... if they saw it on Facebook and then it was repeated 29 times, it had to be true, right? We can educate our community on how to do better than that” (Participant 12)•“I told people ‘This is put out by the CDC. Do not believe the news reporter. Do not believe your best friend on Facebook. They’re talking about conspiracy theories. [Deciding to get vaccinated] is basically on perception” (Participant 4)•“There are no easy solutions to the problem [of getting vaccinated] unless we do away with the internet. Really, I mean then we take out a big chunk of the problem right there especially with social media” (Participant 7)

Trust among close friends and family also provided another type of social pressure that influenced people in making the decision to be vaccinated.
•“On a lot of tough topics, they are not necessarily going to trust outsiders, they’re going to trust their spouse, or their kid, or their sister” (Participant 19)•“They’ve stood their ground. They’ve stood up for their friends who were not going to get vaccinated” (Participant 20)

#### 3.3.3. Barrier 7: Outwardly Visible Breakdown of Trust in Core Community Institutions and Leadership

Key informants from rural communities also expressed a loss of trust between community members and community leaders as a result of the challenges of the COVID-19 pandemic. The breakdown of trust in community leaders resulted in hesitancy to be vaccinated. Conversely, participants identified that where public health entities and local organizations had built more trust with people living in rural communities, there was a more robust uptake of COVID-19 vaccination. Additionally, if trusted members of the community chose to be vaccinated, others in the community would see them as an example and may be more willing to be vaccinated as well.
•“I think that that relationships and trust were already broken [from the spread of COVID-19]. We were reacting to COVID rather than being proactive. It was the breaking of trust and the breaking of relationships between leaders and the community” (Participant 20)•“There are so many ways, even though it is research based, that [community leaders] can present information. But we always have a bit of ourselves that even tends to influence how we look at that research-based information. And then how do we remove our bias and then help other folks to know that we’ve removed our bias to then provide them the facts?” (Participant 13)•“I think a lot of it goes back to relationships and people you trust. In a small town of less than 3000 people, everyone pretty much knows everyone. If they’ve learned people’s choice [to be vaccinated] that might get them to listen and seek them out for advice” (Participant 8)•“My personal take is that word-of-mouth was probably the only thing that worked. At that time, the state and even the county was becoming very political polarized. If they heard it from a neighbor and someone they trusted, they’d get it” (Participant 17)

### 3.4. Perceived Behavioral Control

The construct of perceived behavioral control describes the ability or difficulty an individual believes they will have in performing the behavior. This construct also takes into consideration the amount of autonomy an individual has available to them to make a decision about performing the behavior, in this case the decision to be vaccinated against COVID-19.

#### 3.4.1. Barrier 8: Making Vaccination Against COVID-19 Mandatory Led to Feelings of Lost Autonomy and Reactionary Refusal to Be Vaccinated

Extension staff described observing feelings of protest among people in rural communities with mandates that vaccination was required rather than optional. Perceived behavioral control was negatively impacted, since individuals felt that they no longer had control of this facet of their own health. Interview participants believed people living in rural areas deeply value independence and autonomy. Feeling that their values were being violated substantially reduced interest in getting vaccinated, and in some cases led to reactionary refusal to be vaccinated against COVID-19.
•“There was a stronger feeling of independence and free speech. They were not interested in being told by government what they should do” (Participant 21)•“It doesn’t help when you’re told you must comply. People don’t want to. They’ll comply if they agree with it or see it benefiting themselves. They didn’t see that right off the bat, so there was a lot of independent thinking. [The vaccine] was kind of pushed on them and they ended up pushing back” (Participant 18)•“Well, there was a lot of controversy over [COVID vaccination] because in a rural county people tend to be independent and they and they tended to say, ‘why do we have to do this?’” (Participant 6)•“People just decided that the government wasn’t going to tell them what to do and weren’t vaccinated” (Participant 2)

#### 3.4.2. Barrier 9: Not Enough Information Was Made Available About the Vaccine Technology and Safety for People to Feel Comfortable Being Vaccinated

Another barrier identified by Extension staff interview participants was the concern among rural populations that there was not enough information available about the rigor involved in the mRNA vaccine development process.
•“Some of it was just [wanting to] understand a little bit more about the mRNA technology, or the difference between the different shots, or how the actual approval went. It terrified people that it got emergency approved so quickly” (Participant 19)•“Where we live people have a strong sense of self-sufficiency and want to know the full picture before they jump into something, really wanting to have a good understanding of the findings. In those early days when [vaccine approval] was moving so fast, they weren’t confident about the vaccination what it would do health-wise” (Participant 11)•“I think there are some safety concerns from quite a few individuals. Can this vaccine be developed in such a short period of time and are we cutting corners?” (Participant 8)

Others expressed concerns about mRNA vaccine testing and whether the approval process was carried out to the same standards involved in previous vaccine trials. Informants expressed the viewpoint that more people living in local communities would have been willing to be vaccinated if the vaccine used technology that was already commonplace.
•“People felt like guinea pigs because mRNA was basically one step out of experimental. [Vaccine development] turned around so fast and that scared a lot of people” (Participant 6)•“There were conspiracy theories about [the COVID-19 vaccine] was going to give everybody autism and all kinds of stuff. All that was prevalent here just like in many other places in the country. We’re a conservative rural community in general” (Participant 1)

## 4. Discussion

According to the ToPB, intentions capture motivational factors that influence a behavior and are an indication of how much effort individuals are willing to exert to perform (or avoid) a behavior [21]. This research sought to identify motivational factors among rural community inhabitants that influenced their decision regarding whether to become vaccinated against COVID-19. A stronger intention to engage in a behavior is more likely to result in the performance of that behavior [21]. Motivational factors against vaccination that were identified in this work could be considered and addressed during future public health efforts to provide outreach resulting in greater success in rural communities.

Key informants observed widespread feelings of anger and frustration over vaccine mandates and loss of autonomy. Four of the barriers identified (Barriers 1, 2, 5, and 8) were factors that appeared to be unique to rural areas. In Washington State, people were at risk of losing employment or access to local services and events if they were unvaccinated [24]. For people from rural areas, these feelings were intensified by the belief that vaccination was less important or not necessary at all compared with those living in dense urban areas because of the inherent geographical distancing between individual residences and communities in general. The outdoor nature of many sources of employment in rural communities, such as farming, forestry, and fishing was also a factor because these professions are often socially distanced by nature. ToPB describes that expression of a behavioral intention can only exist if the behavior itself is under the control of the individual [21]. Feeling that vaccination was not needed combined with frustration over vaccination mandates damaged perceived behavioral control and led, in many cases, to refusal because of deeply held values and principles of independence rather than direct concerns about the vaccine itself. Taking a more tailored educational and informational approach, rather than enforcing mandates, may be a more successful approach in future vaccination endeavors.

Previous research also reported general vaccine hesitancy as a significant factor limiting uptake of COVID-19 vaccines in rural communities in other areas of the U.S. [25], although a detailed rationale for the beliefs behind the hesitancy was not included. These results are similar to Barrier 6 in this study, which identified pressures from social media, political parties, and community members as social norms that created a barrier in the decision to get vaccinated. Barrier 7, a breakdown in trust among core community institutions and leadership, could also have had a substantial impact on general community vaccine hesitancy. These results also align with the information provided in the updated CDC Field Guide and State Strategies to Increase COVID-19 Vaccine Uptake in Rural Communities, a document released by the National Governors Association in 2021 [4]. Recommendations from these documents encourage trusted messengers, vaccine champions, and strong messaging from healthcare providers to be involved in local messaging [4,26]. Communicating directly with local community leaders and trusted healthcare entities rather than a top-down dissemination of information directly from state and federal leaders may be beneficial in overcoming these barriers.

Three of the factors identified, Barriers 3, 4, and 9, were related to a lack of available information and transparency about the COVID-19 vaccination development process, mRNA vaccine technology, and efficacy of the vaccine. Similar results about the rapid vaccine development and concerns about vaccine complications have been identified in other research [27]. Concerns about safety of the vaccine and trust in vaccine development among rural populations have also previously been identified [27,28]. Although previous research has attributed such concerns to lower educational attainment [29], the results of the current study suggest that a lack of available information, rather than a lack of general education, was responsible. Future efforts in educational outreach to provide more detailed safety and development information may aid public health officials in improving COVID-19 vaccine uptake in rural areas. Providing this information in all languages that are spoken regionally as well as through channels that do not involve technology would be beneficial in making sure that everyone has enough information to make an informed decision about being vaccinated.

Results of this research should be interpreted within the context of its limitations. Extension staff provided insight from someone working and typically also living in a rural community. They were ideal informants because of their high level of community engagement and understanding of the needs and beliefs of people living nearby. However, their perspective was one “voice” from their community and may not be representative of everyone living in the community. The researchers asked participants to reflect on their observations within the community rather than personal opinions, but participant observations may be influenced by their own personal experiences during the pandemic. Care was also taken by the researchers to ensure they had no prior relationship with the participants they were interviewing. However, there is the possibility that Extension staff were uncomfortable with being involved in research conducted by members of their own university. There is also the potential for selection bias, since interviews were conducted with people from Extension who were willing to participate. Some Extension staff who had strong negative opinions on this topic or felt this topic was too controversial may have chosen not to participate. This research also has the potential for recall bias, since informants were asked to consider events that took place up to three years prior to the conversation.

Representation from rural communities in one U.S. state may not be generalizable to other states or countries. Some similarities among studies evaluating COVID-19 vaccination barriers in other rural areas of the United States exist [25,27,28], but future research evaluating additional geographic areas would also be beneficial. Addressing barriers to COVID-19 vaccination uptake in rural areas could be beneficial in future public health vaccination efforts.

## 5. Conclusions

This study was designed to identify vaccination barriers in rural areas using the constructs of the ToPB. Nine unique barriers were identified which influenced the decision to receive the COVID-19 vaccination among people living in rural areas in Washington State. Barriers included inherent social distancing related to rural location negating vaccine necessity; lack of early vaccine availability in rural locales; concerns about the effectiveness of vaccine; inadequate dissemination of vaccine information to non-native English language speakers and those with limited access to technology; perception of exclusion of rural populations’ unique needs during the design and implementation of vaccine mandates; exertion of social pressures on rural individuals’ vaccine uptake decision; highly visible breakdown in standard trust in core community institutions and leadership; vaccine mandates leading to a self-perceived loss of autonomy; and limitations in vaccine technology information influencing the perception of vaccine safety. Barriers identified in this study illustrate the importance of collaborating with community liaisons and trusted community members during unprecedented events. By identifying barriers from the previous pandemic, future outreach efforts can be designed to improve intention and vaccination uptake.

## Figures and Tables

**Figure 1 vaccines-12-01442-f001:**
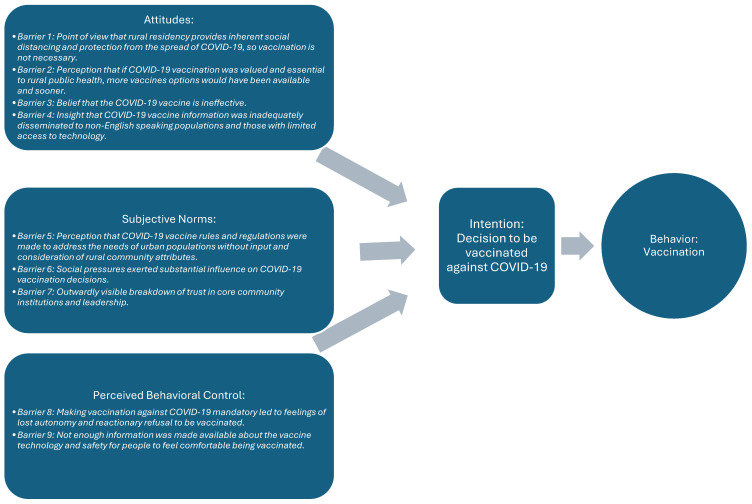
Project results organized using the constructs of the ToPB [21].

**Table 1 vaccines-12-01442-t001:** Demographics and representation of Extension staff participants.

Demographic	Average	Range
Number of years working at WSU Extension	16.2 years	2.5 to 53 years
Number of years working at current site	13 years	1 to 53 years
Number of counties served by Extension offices	1.4 counties	1 to 3
Percentage of county/reservation classified as rural	89% rural	25% to 100%
Participant gender	57% Female (n = 12) 43% Male (n = 9)	not applicable

## Data Availability

The datasets presented in this article are not readily available because the data are part of an ongoing study. Requests to access the datasets should be directed to the corresponding author.

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
