# Peer review of "A Qualitative Analysis of Rural Community Vaccination Barriers During the COVID-19 Pandemic"

_vaccines, 2024, doi:10.3390/vaccines12121442_

Round 1
Reviewer 1 Report
Comments and Suggestions for Authors
This small-scale interview tried to identify barriers to COVID-19 vaccine among patients living in rural areas in USA. But for readers out of USA, they need information on the geography of Washington state which I think it should be included. The authors need to provide more background information about Washington state like the geography, economy, population size etc. so readers outside USA can understand the research setting.
The demographics of the participants seem incomplete, and I need to know much more and see if the study finding is valid. What is needed includes age, co-morbidities, education level etc. Please provide the relevant data in main text or tables.
Lack of graphical illustration is another weakness and makes this article difficult to read. I will suggest the authors present the data with appropriate figures and tables to increase the readability of the manuscript.
Author Response
Thank you for taking the time to provide this thoughtful review of our manuscript! Our responses to your comments and those from the editor and other reviewers are included in the attached document.

Reviewer 2 Report
Comments and Suggestions for Authors
Kimberly McKeirnan investigate in their manuscript the reasons why people from a rural environment in Washington State, in the US, have a higher barrier to get vaccinated during the COVID-19 pandemic.
My resumé of reviewing this article is that there is very little I can propose to improve the manuscript. It is very well structured, methods are clearly described and the results very well presented.
Thee authors interpreted the results within the Theory of Planned Behaviour framework and identified nine different distinct barriers. Knowing that these barriers exist allow to prepare a potential future crisis to improve uptake of a vaccine.
The authors, coming themselves from the Washington State University choose a unique way to contact people from their rural surrounding via University Extension staff - established since many years - to develop relationships between Universities and the rural population.
The development of the interview script is well described. It contains open-ended questions which is a pre-requisite to allow for surprising results.
The interviews were extended until the interviewer concluded that a saturation point was reached - no new arguments were brought forward in the interviews.
The analysis of the interview results is well described by projecting them onto the theoretical framework of the Theory of Planned Behaviour.
Figure 1 summarises all the results extracted from the interviews in a condensed form which are well represented by individual directly cited statements in the article. In the discussion the authors make recommendations how to increase the vaccination uptake in rural communities considering the identified barriers. They end it by mentioning the limitations of their investigation.
I think the article can be published as it is.
Author Response

(The authors gave the same response as above.)

Reviewer 3 Report
Comments and Suggestions for Authors
This is an important and well conducted study, using Extension workers with intimate knowledge of the districts they serve. It provides important guidance to improving the information provided to population groups and addressing their key concerns.
My comments are addressed to improving the data in tehri study to guide users and future researchers.
“CDC report using a social vulnerability index identified 15 fac-56 tors such as poverty and lack of transportation that decreased community capacity to 57 respond in an emergency [5]. Disparities in county-level vaccination coverage by social 58 vulnerability have increased as vaccine eligibility has expanded, especially in areas out-59 side of large cities [5]. Vaccination coverage among adults is also reduced among people 60 in counties with lower socioeconomic status and with higher percentages of households 61 with children, single parents, and persons with disabilities [6].”
[can you provide % data please to show how large the problem is?
“Additionally, un-68 derrepresentation of minority groups in the COVID-19 vaccine trials was found to con-69 tribute to vaccine hesitancy and reduced uptake among minority populations [8].”
“The Kaiser Family Foundation reports that rural residents are 78 among the most vaccine hesitant groups, along with Republicans, individuals 30-49 79 years old, and Black adults [10].”
[Again, can you provide % data for each of the statements that you make so that future researchers will know how to plan theri studies?
Methods
“semi-structured key informant interview was developed, including a combination 158 of open ended and close-ended questions. Closed-ended questions were used to gather 159 demographic data. The open-ended questions were utilized to elicit more detailed and 160 thoughtful responses regarding COVID-19 vaccination barriers. A semi-structured format 161 was chosen so that the interviewer would use the script as a guide while having the free-162 dom of including probing questions when needed to dive into topics deeper. Key inform-163 ant interview methodology allows for variation from the script to add probing questions 164 as needed to gain additional insight and clarification on an individual basis.”
“semi-structured key informant interview was developed, including a combination 158 of open ended and close-ended questions. Closed-ended questions were used to gather 159 demographic data. The open-ended questions were utilized to elicit more detailed and 160 thoughtful responses regarding COVID-19 vaccination barriers. A semi-structured format 161 was chosen so that the interviewer would use the script as a guide while having the free-162 dom of including probing questions when needed to dive into topics deeper. Key inform-163 ant interview methodology allows for variation from the script to add probing questions 164 as needed to gain additional insight and clarification on an individual basis.”
How much initial disagreement was there and how was this resolved?
Results
“Another factor reported by participants that contributed to vaccine hesitancy was lim-309 ited information. Participants reported observing that at times information was not com-310 municated in a way that could be understood by people who did not speak English or did 311 not have ready access to a computer. 312
• “There was very limited information for Spanish-speaking people. It was terrible. They 313 were frightened. They didn’t know what they were getting into, what was happening.” 314 (Participants 2) 315
• “Among some of the poorer economic demographics in our community, I saw that 316 they were not getting enough information to make informed decisions so that they just 317 weren't making decisions. A lot of folks said, ‘I don't know what to do, so I'm going to 318 do nothing’.” (Participant 9) 319
• “The cultures that are not the dominant cultures anywhere have a distrust or hesitancy 320 to trust the mainstream culture, government, medicine, everything that they see. So I 321 would guess that the minority populations in any county were less likely to be vac-322 cinated than the mainstream culture.” (Participant 20) ”
[Can you provide data on what information in appropriate languages was provided to these residents, perhaps with quotes that describe vaccine efficacy and adverse effects]
Author Response

(The authors gave the same response as above.)
